# 3-[5-(1*H*-Indol-3-ylmethylene)-4-oxo-2-thioxothiazolidin-3-yl]-propionic Acid as a Potential Polypharmacological Agent

Yulian Konechnyi [1,*] , Andrii Lozynskyi [2], Iryna Ivasechko [3], Tetiana Dumych [3], Solomiya Paryzhak [3], Oksana Hrushka [4], Ulyana Partyka [5], Iryna Pasichnyuk [6], Dmytro Khylyuk [7] and Roman Lesyk [2,*]

1 Department of Microbiology, Danylo Halytsky Lviv National Medical University, 69 Pekarska, 79010 Lviv, Ukraine
2 Department of Pharmaceutical, Organic and Bioorganic Chemistry, Danylo Halytsky Lviv National Medical University, 69 Pekarska, 79010 Lviv, Ukraine
3 Department of Regulation of Cell Proliferation and Apoptosis, Institute of Cell Biology, Position at the NAS of Ukraine, 79000 Lviv, Ukraine
4 Central Research Laboratory and Industrial Toxicology Laboratory, Danylo Halytsky Lviv National Medical University, Pekarska Str., 69 a, 79010 Lviv, Ukraine
5 Andrej Krypynsky Lviv Medical Academy, Department of Pharmacy, 79010 Lviv, Ukraine
6 Department of Pediatrics No.1, Medical Faculty, Danylo Halytsky Lviv National Medical University, 79010 Lviv, Ukraine
7 Department of Organic Chemistry, Faculty of Pharmacy, Medical University of Lublin, 4A Chodźki Street, 20-093 Lublin, Poland
* Correspondence: yuliankonechnyi@gmail.com (Y.K.); dr_r_lesyk@org.lviv.net (R.L.); Tel.: +380-(032)-275-59-66 (R.L.)

**Abstract:** Searching for new types of biological activities among preliminarily identified hit compounds is a key challenge in modern medicinal chemistry. In our study, a previously studied 3-[5-(1*H*-indol-3-ylmethylene)-4-oxo-2-thioxothiazolidin-3-yl]-propionic acid (Les-6614) was screened for antimicrobial, antifungal, anti-allergic, and antitumor activities. Moreover, cytotoxicity, molecular docking, and SwissAdme online target screening were accomplished. It was determined that the Les-6614 has slight antimicrobial and antitumor activity. However, the studied compound decreased IgE levels in sensitized guinea pigs by 33–86% and reduced IgA, IgM, IL-2, and TNF-$\alpha$, indicating anti-inflammatory and anti-allergic aactivities. According to the SwissADME web tool, target predictions for Les-6614 potentially have an affinity for lysosomal protective protein, Thromboxane-A synthase, and PPAR$\gamma$. The molecular docking confirmed that the studied 2-thioxo-4-thiazolidinone derivative showed good bonding with LLP and TXAS, leading to stable protein–ligand complexes. Additionally, Les-6614 is a potential PPAR$\gamma$ modulator, which is important in the pathogenesis of allergy, cancer, and cardiovascular diseases.

**Keywords:** 2-thioxo-4-thiazolidinones; antimicrobial activity; guinea pigs; inflammation; toxicity; molecular docking; PPAR-$\gamma$

## 1. Introduction

One approach to the search for new drugs is to select the active molecule as a polypharmacological agent with polytargeted action, based on the concept of "single agent on multiple targets for single/multiple disease/es" [1]. The absence of a pronounced effect on one target does not always mean the molecule is unpromising, but the number of new investigated targets increases yearly. Often, the remoteness of the pronounced effect of the active molecule on the target can indicate low toxicity and the possibility of prolonged action.

In searching for new hit compounds and drug candidates, the unification of the approach to the screening process remains problematic [2]. A small list of targets is often considered, which speeds up selecting the most active molecule, but small specific targets are rejected. On the other hand, involving many biological targets in the screening process significantly increases the cost of screening, often without increasing efficiency. Finding

the "golden mean" in optimizing the screening of biologically active molecules remains relevant for big pharma and small scientific pharmaceutical groups [3].

This manuscript presents one selected molecule tested by different methods against different targets and is an example of a weakly active derivative that does not have a single pronounced action but is still promising for further research. In this study, we explored preliminary synthesized 2-thioxo-4-thiazolidinone (rhodanine) derivative Les-6614 (Figure 1) as a potential agent with antimicrobial/antifungal/anti-allergic/anti-inflammatory/antitumor effects.

**Figure 1.** Structure of 3-[5-(1*H*-indol-3-ylmethylene)-4-oxo-2-thioxo-thiazolidin-3-yl]-propionic acid (Les-6614).

In our previous work, we described the synthesis of 5-indolylmethylene-3-substituted rhodanines via Knoevenagel reaction of rhodanine-3-propanoic/ethanesulfonic acids and their esters with indole-3/6-carbaldehydes [4,5]. Among the synthesized compounds, the highest activity showed 3-[5-(1*H*-indol-3-ylmethylene)-4-oxo-2-thioxothiazolidin-3-yl]-propionic acid (Les-6614). Previously, we found that the mentioned compound has low toxicity, low/moderate antifungal, and immunomodulatory activities (increase in the quantitative level of lymphocytes and monocytes against the background of preservation of the number of neutrophilic leukocytes, increased absorption activity of phagocytes is accompanied by a simultaneous decrease in the level of completion of phagocytosis, an increase in the number of CD3 and CD8 T-lymphocytes against the background of a reduction in the relative number of natural killer cells and B-lymphocytes CD22).

## 2. Materials and Methods

### 2.1. Antimicrobial Activity

The studied derivative Les-6614 was tested in vitro for its antibacterial and antifungal activities using agar diffusion and serial dilutions resazurin-based microdilution assays (RBMA) [6,7]. For this purpose, 100 μL (1 mg/mL) of the tested compound was placed in an agar well with a diameter of 5.5 mm. The diameter of the growth retardation was measured using a micrometer with an error of 0.1 mm. Dimethyl sulfoxide (DMSO) (Sigma-Aldrich, St. Louis, MO, USA), vancomycin (discs) (Thermo Fisher Scientific Inc., Waltham, MA, USA), ciprofloxacin (discs) (Thermo Fisher Scientific Inc., Waltham, MA, USA), and clotrimazole (discs) (Mast Group Ltd., Liverpool, UK) were used as controls. Pure DMSO was used as a solvent due to the poor solubility of the test compound in dilute DMSO. In addition, Mueller-Hinton agar (Sigma-Aldrich, St. Louis, MO, USA) and Saburo agar (for fungi) (Sigma-Aldrich, St. Louis, MO, USA) were used, and Petri dishes were incubated at 37 °C for 24 h for bacteria and at 25 °C for 24–48 h for fungi. The RBMA method involved the introduction of a 96-well plate of 50 μL of nutrient medium (Mueller-Hinton broth or glucose broth), 50 mL of a suspension of the microorganism (McFarland 1.0–1.5), and 100 μL of the tested compound, with the addition of 15 μL of 0.02% resazurin in each well. Twenty-three reference and clinical microbial and fungal strains were used (Table 1), previously identified by the MALDI TOF system (Bruker, Bremen, Germany) and 16S rRNA gene sequences. All clinical strains were multidrug-resistant or extensively drug-resistant with

different antibiotic resistance patterns. Clinical strains were isolated from a patient with healthcare-associated infections (respiratory tract, blood, urine) from regional hospitals. All testing was repeated triplicate.

**Table 1.** In vitro antimicrobial activity of Les-6614 (zone of growth inhibition at conc. 1 mg/mL after 24–48 h and MIC value μM).

| N | Type of Species | Species of Bacteria and Fungi | Zone of Growth Inhibition (mm ± SE) and Some MIC Value (μM) | | | | |
|---|---|---|---|---|---|---|---|
| | | | Les-6614 | DMSO | Vanco-Mycin | Cipro-Floxacin | Clotri-Mazole |
| 1 | Gram-negative bacteria | *Pseudomonas aeruginosa* ATCC 10145 | 7.0 ± 0.3 | 7.0 ± 0.3 | - | 35.0 ± 0.3 | - |
| 2 | | *Raoultella terrigena* ATCC 33257 | 6.5 ± 0.25 | 6.5 ± 0.25 | - | 42.0 ± 0.5 | - |
| 3 | | *Raoultella ornithinolytica* DSM 7464 | 7.0 ± 0.3 | 7.0 ± 0.3 | - | 40.0 ± 0.5 | - |
| 4 | | *Escherichia coli* ATCC 25922 | n/a | n/a | - | 28.0 ± 0.5 | - |
| 5 | | *Klebsiella pneumoniae* 189 | n/a | n/a | - | 20.0 ± 0.2 | - |
| 6 | | *Aeromonas hydrophila* N196 | 8.0 ± 0.25 | n/a | - | 27.0 ± 0.4 | - |
| 7 | | *Escherichia coli* N168 | n/a | n/a | - | 15.3 ± 0.2 | - |
| 8 | | *Morganella morganii* N55 | 7.0 ± 0.3 | 7.0 ± 0.3 | - | 21.0 ± 0.4 | - |
| 9 | | *Kocuria marina* N133 | 7.0 ± 0.3 | 7.0 ± 0.3 | - | 27.5 ± 0.5 | - |
| 10 | Gram-positive bacteria | *Streptococcus agalactiae* ATCC 13813 | 8.8 ± 0.25 | n/a | 32.0 ± 0.5 | - | - |
| 11 | | *Staphylococcus aureus subsp. aureus* ATCC 25923 | 9.4 ± 0.4 | n/a | 32.0 ± 0.5 | 35.0 ± 0.5 | - |
| 12 | | *Staphylococcus epidermidis* ATCC 12228 | 9.5 ± 0.5 | 12.0 ± 0.5 | 25.0 ± 0.5 | 52.5 ± 0.8 | - |
| 13 | | *Baccilus subtilis* ATCC 6633 | 9.4 ± 0.2 | 7.4 ± 0.2 | 27.0 ± 0.5 | - | - |
| 14 | | *Staphylococcus aureus* N 23 | 8.2 ± 0.4 | n/a | 11.4 ± 0.3 | 9.0 ± 0.2 | - |
| 15 | | *Enterococcus faecalis* N26 | n/a | n/a | 11.1 ± 0.3 | 13.3 ± 0.5 | - |
| 16 | | *Enterococcus faecalis* N191 | n/a | n/a | 11.0 ± 0.4 | 13.4 ± 0.5 | - |
| 17 | Fungi | *Candida. albicans* (ATCC 885-653) | 9.0 ± 0.4 | n/a | - | - | 18.0 ± 0.5 |
| 18 | | *Candida albicans* N67 | 9.0 ± 0.3/MIC 3003 μM | n/a | - | - | 11.0 ± 0.3 MIC 2.9 μM |
| 19 | | *Candida guilliermondii* N83 | n/a | n/a | - | - | 11.0 ± 0.3 |
| 20 | | *Candida utilis* N127 | n/a | n/a | - | - | 11.0 ± 0.3 |
| 21 | | *Candida kefyr* N92 | n/a | n/a | - | - | 11.0 ± 0.3 |
| 22 | | *Candida lusitaniae* N89 | 7.0 ± 0.3 | n/a | - | - | 11.0 ± 0.3 |
| 23 | | *Saccharomyces cerevisiae* N62 | 7.0 ± 0.4 | n/a | - | - | 8.0 ± 0.2 |

Vancomycin 30 μg (inhibition zone 17–21 mm for *S. aureus*); ciprofloxacin 5 μg (inhibition zone 25–33 mm for *P. aeruginosa*, 22–30 mm for *S. aureus*, 30–40 mm for *E. coli*); clotrimazole 10 μg (inhibition zone 12–17 mm for *Candida* spp.; dameter of well 5.5 mm; n/a no activity.

### 2.2. Cytotoxicity and Antitumor Effect In Vitro Cell Lines

Cell lines HCT-116 wt, MCF-7, KB3-1, K562, J774.2, NIH 3T3, and HaCaT were kindly provided by a Collection at the Institute of Molecular Biology and Genetics, National Academy of Sciences of Ukraine (Kyiv, Ukraine). The p53-deficient HCT-116 p53 (-/-) colon cancer cells were donated by a Collection of the Institute for Cancer Research at Vienna Medical University (Vienna, Austria). Cells were cultivated in Dulbecco's Modified Eagle Medium (DMEM, Biowest, Nuaillé, France) or RPMI-1640 medium (Biowest, France), containing 10% of fetal bovine serum (FBS, Biowest, Nuaillé, France) according to the recommendations of American Type Culture Collection (ATCC), under the incubation condition of 5% $CO_2$ humidity at 37 °C.

*MTT assay for measuring cells viability*

The cytotoxic action of the compound was evaluated using MTT (3-[4,5-dimethylthiazol-2-yl]-2,5-diphenyl tetrazolium bromide) assay (Sigma-Aldrich, St. Louis, MO, USA). The 4000 adherent or 15,000 suspension cells per well were seeded in 96-well plates in 100 μL DMEM or RPMI-1640, supplemented with 10% heat-inactivated fetal bovine serum, and

incubated for 72 h at 37 °C in $CO_2$-incubator with studied compound at final concentrations of 1, 10, and 100 μM. After incubation, the medium was replaced with MTT reagent (5 mg/mL) and incubated for the next 4 h. Crystals of formazan were dissolved in the dimethylsulfoxide, and the reaction absorbance was measured by an Absorbance Reader BioTek ELx800 (BioTek Instruments Inc., Winooski, VT, USA). Using nonlinear regression, the half maximal inhibitory concentration value ($IC_{50}$) was calculated by GraphPad Prism 8 v. 8.0.1 software (GraphPad Software, San Diego, CA, USA). The results were analyzed with GraphPad Prism 8 v. 8.0.1 (GraphPad Software, San Diego, CA, USA) and presented as a mean (M) $\pm$ standard deviation (SD) of three parallels [8].

### 2.3. Animals and Immunological (ELISA) Studies

Studies of the allergenic effect of the compound were performed according to the guidelines [9,10]. Sensitization was performed by intradermal injection of 200 μg of the derivative diluted in 0.02 mL of solvent (saline for the control group of pure and 20% solution of ethyl alcohol) in the skin of the outer surface of the ear animals. Control animals were injected with 0.02 mL of solvent. After 10 days, an additional 20 applications were applied to the pre-depilated area of the lateral surface of the body. The degree of sensitization was established after intradermal testing in dilutions: 1:10, 1:100 [11–13]. The Immunological studies were performed using erythrocyte diagnosticums (manufactured by TOV NVL "Granum", Kharkiv, Ukraine) and ELISA kits, guinea pig interleukin 6, guinea pig IgG, guinea pig IgA, guinea pig IgM, and guinea pig tumor necrosis factor alpha, per 48 tests (MyBioSource, Inc., San Diego, CA, USA).

### 2.4. Molecular and Pharmacokinetic Properties

The physical properties and adsorption, distribution, metabolism, elimination, and toxicity (ADMET) parameters of (*Z*)-3-(5-((1*H*-indol-3-yl)methylene)-4-oxo-2-thioxothiazolidin-3-yl)propanoic acid was calculated using the SwissAdme online server of the Swiss Institute of Bioinformatics [http://www.swissadme.ch/index.php accessed on 20 November 2022].

### 2.5. Molecular Docking

The molecule's structure was drawn using ChemDraw v.18.0.0.231 software (PerkinElmer, Waltham, MA, USA), and energy was minimized using Avogadro (freely available software under an open-source license), employing molecular mechanics with the force field MMFF94 [14], and protonation of the ligand was performed according to pH 7.4.

As target enzymes, we chose the lysosomal protective protein LPP (PDB code 4CIA), PPARγ-receptors (PDB code 5Y2O), vascular endothelial growth factor receptor 2 VEGFR2 (PDB code 4AGD), and thromboxane A2 synthase (TXAS). VEGFR2 was chosen owing to the structural similarity of the tested compound to sunitinib, a multi-targeted receptor tyrosine kinase inhibitor. Because of the absent three-dimensional structure of the thromboxane A2 synthase, homology modeling was performed using the SwissADME portal (https://swissmodel.expasy.org/ accessed on 20 November 2022). The GMQE and QME-ANDisCo scores for the obtained model were 0.67 and 0.70 $\pm$ 0.05, respectively, considering the obtained model possesses good quality. As the reference agents, the co-crystalized compounds with the explored enzymes (LPP, PPARγ, VEGFR2) and dazoxiben as reference inhibitors for thromboxane A2 synthase were used [15]. The dimensions of the grids for three proteins docking were set as 60 · 60 · 60 Å, with a spacing of 0.375 Å between the grid points. AutoDock4 tools with the implementation of the Lamarckian genetic search algorithm (LGA) [16] were chosen for all docking simulations. The optimized AutoDocking run parameters are as follows: the number of GA runs was increased to 50, and population sizes were set as 300. All other run parameters were maintained at their default settings. The RMSDs values were lower ($\leq 2$ Å), which were indicated consequently by parameters for the docking simulation, which are reasonable in reproducing the positions of the initial ligands from the X-ray crystal structures and can be extended to search the enzyme binding conformations for other chemical entities. The protein–ligand interactions, such

as hydrogen bonding and other non-bonded energies between tested compound with the selected enzymes for simulation were visualized using Biovia Discovery Studio Visualizer v. 21.1.0.20298 software (Dassault Systemes, Vélizy-Villacoublay, France).

*2.6. Statistical Analysis*

Statistical data processing was performed using Microsoft Excel software. Data were presented as arithmetic mean (M) and standard deviation (m). The reliability of the obtained data was evaluated using Student's t-test. The level of statistical significance was considered as $p < 0.05$.

**3. Results**

*3.1. Antimicrobial Activity*

Based on the preliminary results of antimicrobial activity 3-[5-(1*H*-indol-3-ylmethylene)-4-oxo-2-thioxothiazolidin-3-yl]-propionic acid and rhodanine derivatives in general [17,18], we conducted further studies on additional strains of microorganisms. The tested compound showed weak antimicrobial activity against some Gram-positive and Gram-negative microorganisms, as well as weak/moderate antifungal activity against yeast fungi (MIC 3003 μM (25 μg/mL) against *Candida albicans* 67) (Table 1).

*3.2. Cytotoxicity and Antitumor Effect In Vitro*

The studied compound Les-6614 was also evaluated for its antitumor potential toward several tumor cell lines, including colon (HCT-116, HCT-116 p53-/-), breast (MCF-7), leukemia (K562), and cervix (KB3-1) lines. This molecule did not significantly influence the viability of all studied cancer cell lines. The $IC_{50}$ of Les-6614 for the most sensitive cell line K562 was 83.20 μM, and, for other lines, it was more than 100 μM (Table 2).

**Table 2.** $IC_{50}$ value of Les-6614 for a panel of human and animal cell lines (MTT test, 72 h, m $\pm$ SD, μM).

| Comp./Cell Line | HCT-116 | HCT-116 p53-/- | MCF-7 | KB3-1 | K562 | J774.2 | NIH 3T3 | HaCaT |
|---|---|---|---|---|---|---|---|---|
| Les-6614 | >100 | >100 | >100 | >100 | 83.20 $\pm$ 2.25 | 98.00 $\pm$ 2.30 | >100 | >100 |
| Doxorubicin | 0.57 $\pm$ 1.22 | 1.36 $\pm$ 0.91 | 1.53 $\pm$ 1.40 | 1.20 $\pm$ 1.38 | 1.34 $\pm$ 0.4 | 1.20 $\pm$ 0.53 | 1.56 $\pm$ 0.22 | >10 |

The cytotoxic action of this compound also was investigated on murine macrophages of the J774.2 line, normal mouse embryonic fibroblasts of the NIH 3T3 line, and human epidermal keratinocytes of the HaCaT line. The strongest impact of this compound was toward J774.2 cells, and $IC_{50}$ was 98 μM (Table 2). It can be assumed that Les-6614 did not show a significant antitumor effect. Its activity did not affect the viability of tumor and pseudo-normal human and animal cell lines compared to doxorubicin ($IC_{50}$ was from 0.57 to >10 μM), which was used as the positive control and is well known anticancer agent.

*3.3. Immunological (ELISA) Studies*

The effect of the tested compound on the level of immunoglobulins and interleukins in the serum of laboratory animals showed a significant decrease in IgA by 21.4% ($p < 0.05$) and a significant increase in IgG by 21.2%. The levels of other immunoglobulins (Table 3) decreased, and the IgE level decreased most significantly by 86.8%.

The results of the cytokine profile showed a slight decrease in the amount of IL-2 and TNF-$\alpha$, which are produced by Th1 and stimulate the processes of cellular immunity. Some elevation of IL-6 will instead regulate the immune response.

*3.4. Molecular and Pharmacokinetic Properties*

ADME prediction of the tested compound was determined using the SwissAdme online server and suggested that the mentioned compound is suited to Lipinski's rule of five, may exhibit high gastrointestinal absorption, and it does not cross the blood–brain barrier. The predicted lipophilicity of the studied derivative given by log Po/w

revealed good permeability and oral absorption through the cell membrane. The tested compound was not to be P-glycoprotein substrates, suggesting that they could not be associated with the excretion of the drug. The negative skin permeability of the tested compound may exhibit low skin permeation across the cell membrane. On the other hand, the SwissAdme predicted that the tested compound might have a pharmacokinetic effect on some cytochrome P450 enzymes (CYP450), such as CYP1A2, CYP2C19, CYP2C9, and CYP3A4. All the predictive data allow considering Les-6614 as a prospective drug-like candidate for further in-depth studies (Table 4).

**Table 3.** Quantitative indicators of the levels of immunoglobulins and interleukins in the serum of laboratory animals under the influence of the tested Les-6614.

| Indicator | Control | 25 Per * | 75 Per * | Les-6614 | 25 Per * | 75 Per * | Change % | *p* |
|---|---|---|---|---|---|---|---|---|
| Ig E, ng/mL | 173.7 | 21.60 | 183.80 | 22.9 | <2.3 | 40.80 | −86.82 | |
| Ig E <sup>&</sup>, ng/mL | 244.6 | 200.4 | 302.7 | 161.6 | 103.4 | 144.1 | −33.93 | |
| Ig A, g/L | 1.6 ± 0.2 | | | 1.1 ± 0.1 | | | −35.29 | $p < 0.05$ |
| IgM, g/L | 0.55 ± 0.04 | | | 0.48 ± 0.02 | | | −12.73 | $p > 0.05$ |
| Ig G, g/L | 8.5 ± 0.3 | | | 10.3 ± 0.2 | | | 21.18 | $p < 0.05$ |
| IL-2, pg/mL | 115.7 | 100.7 | 144 | 76.9 | 7.1 | 106 | −33.54 | |
| IL-6, pg/mL | 257.6 | 240.8 | 285.2 | 276.11 | 249.5 | 296.8 | 7.19 | |
| TNF-α, pg/mL | 44.5 | 28.6 | 58.2 | 26.2 | 21.1 | 27.6 | −41.12 | |

* The 25/75 percentile represents incorrect distribution; the correct distribution is represented by Student's *t*-test (*p*). <sup>&</sup> using a kit from another distributor.

**Table 4.** Physicochemical and pharmacokinetics properties of studied Les-6614.

| | *Physicochemical properties* | |
|---|---|---|
| 1 | Molecular weight | 332.40 |
| 2 | Num. heavy atoms | 22 |
| 3 | Num. arom. heavy atoms | 9 |
| 4 | Num. rotatable bonds | 4 |
| 5 | Num. H-bond acceptors | 3 |
| 6 | Num. H-bond donors | 2 |
| 7 | Molar Refractivity | 94.48 |
| 8 | TPSA Å$^2$ | 130.79 |
| 9 | Consensus log Po/w | 2.29 |
| 10 | Lipinski rule | Yes |
| | *Pharmacokinetics* | |
| 11 | GI absorption | High |
| 12 | BBB permeant | No |
| 13 | P-gp substrate | No |
| 14 | CYP1A2 inhibitor | Yes |
| 15 | CYP2C19 inhibitor | Yes |
| 16 | CYP2C9 inhibitor | Yes |
| 17 | CYP2D6 inhibitor | No |
| 18 | CYP3A4 inhibitor | Yes |
| 19 | Log Kp (SP) (cm/s) (skin permeation) | −6.48 |
| 20 | Bioavailability score | 0.56 |

　　　　Continuing our in silico studies, we employed similarity score measures using the SwissSimilarity, which provides several two-dimensional screening methods with FP2 topological chemical fingerprints for finding similar drug candidates or known bioactive molecules available for all libraries of small molecules [19]. The similarity score ranges from 0 for totally different molecules to 1 for identical compounds. Thus, path-based (linear) molecular fingerprint analysis from the DrugBank database identified four compounds with similarity scores ranging from 0.511–0.580. The mentioned compounds have an affinity to several biotargets, namely, the lethal factor of *Bacillus anthracis* (s.s. 0.580), glycylpeptide N-tetradecanoyltransferase 2 (s.s. 0.567), phosphatidylinositol 4,5-bisphosphate 3-kinase

catalytic subunit gamma isoform (s.s. 0.536), and serine/threonine protein kinase (s.s. 0.511) (Figure S1).

Based on previous findings, we were specifically interested in exploring the possible targets of (Z)-3-(5-((1*H*-indol-3-yl)methylene)-4-oxo-2-thioxothiazolidin-3-yl)propanoic acid. The SwissTargetPrediction is an online tool specialized in target prediction based on the structural similarities between investigative compounds and well established ligands [20]. The lysosomal protective protein (UniprotID: P10619, CHEMBL ID: 6115), thromboxane-A synthase (UniprotID: P24557, CHEMBL ID: 1835), and peroxisome proliferator-activated receptor gamma (UniprotID: P37231, CHEMBL ID: 235) were the top predicted targets due to the similarity of tested compound (Figure S2).

### 3.5. Molecular Docking

One effective way to identify promising agents among many synthesized derivatives is to perform docking research, which helps perform the best further changes in the molecule's structures for enhancing the biological activities of the new compounds. Hence, in silico docking simulations are powerful techniques for obtaining significant information for further structure optimizations of the previously identified active agents.

In the present work, molecular docking studies were conducted to determine the molecular binding modes of the investigated compound with potent biological targets. Docking studies were conducted using AutoDock Tool v. 1.5.6. (Scripps Research, San Diego, CA, USA) to determine the free energy, inhibition constant, and virtual binding modes of the possible target compound–enzyme complexes.

Docking simulations suggest moderate affinities to all tested proteins, which confirms the assumption about considering the target compound as the initial scaffold for further design of new chemical entities with the improved pharmacological profile. The results of the simulations are highlighted in Table 5.

**Table 5.** Molecular docking results for Les-6614.

| Compound | LPP (PDB 4CIA) | | PPARγ (PDB 5Y2O) | | VEGFR2 (PDB 4AGD) | | TXAS | |
|---|---|---|---|---|---|---|---|---|
| | Binding Energy | Inhibition Constant, Ki, μM | Binding Energy | Inhibition Constant, Ki, μM | Binding Energy | Inhibition Constant, Ki, μM | Binding Energy | Inhibition Constant, Ki, μM |
| Compound | −8.47 | 0.62 | −8.49 | 0.60 | −9.43 | 0.121 | −9.07 | 0.225 |
| 6KZ [1] | −9.42 | 0.125 | - | - | - | - | - | - |
| Pioglitazone | - | - | −13.12 | 0.00024 | - | - | - | - |
| Sunitinib | - | - | - | - | −12.61 | 0.00057 | - | - |
| Dazoxiben | - | - | - | - | - | - | −10.14 | 0.037 |

[1] 6KZ—PDBe Ligand Code from PDB structure of 4CIA, reported inhibitor of the cathepsin A.

The most promising directions for developing new biologically active agents based on the target compound structure are the new LLP and TXAS inhibitors. Binding energies with the proteins mentioned above are close to the reference ligands, which may confirm the prognosis algorithm of the SwissADME portal. The anticancer activity of the compound may be connected with the low affinity to PPARγ and VEGFR2, but the obtained results do not allow us to make strong suggestions about the mechanism of the anticancer activity.

## 4. Discussion

Our study describes one selected biologically active molecule Les-6614 without a pronounced action toward one separate target. The studied compound demonstrated low antimicrobial and antifungal activity and low antitumor activity in vitro while being non-toxic.

Lysosomal protective protein (UniprotID: P10619) with the active district cathepsin A is one of the predicted compound targets. Cathepsin A, a crucial regulator of galactoside metabolism, has been linked to aggressive behavior and a poor prognosis in breast ductal carcinoma in situ and may be used as a marker to predict the presence of concurrent

invasion in this kind of cancer [21,22]. Lysosomal protective proteins, such as CHEMBL ID: 6115, do not have PubChem Compound Identifiers, but they are easily found with UniprotID [23]. CatA has recently been discovered as a possible therapeutic target for reducing cardiac hypertrophy since it can inactivate endothelin 1, a vasodilator and pro-inflammatory mediator [24].

A new target for treating cardiovascular diseases, thromboxane A synthase (UniprotID: P24557, CHEMBL ID: 1835), interacts with the thromboxane prostanoid receptor, displays biological activity by interacting with a G protein-coupled receptor, causes platelet activation and aggregation, and causes smooth muscle contraction. The tested compound potentially may block thromboxane A synthase and prevent cardiovascular disease (prevent platelet aggregation) [25,26].

Peroxisome proliferator-activated receptor gamma (PPARγ) (UniprotID: P37231, CHEMBL ID: 235) could be a target for diabetes mellitus; also, PPARγ is a key regulator of a variety of cell types implicated in the inflammatory response to allergens, including T helper cells, airway epithelial cells, and breast cancer carcinogenesis [27–30]. A recent review demonstrated a potentially strong connection between PPARγ and the development of allergic diseases (asthma, dermatitis) [31] and identified PPARγ -modulators as potential anti-allergic agents. This is consistent with the results of our study of guinea pigs. This compound led to the reduction in Ig E by 33–86%, Ig A and IgM, and IL-2 and TNF-α, indicating an anti-inflammatory and anti-allergic effect. The role of PPARγ in the pathogenesis of allergy requires further research and the place of allergy in cancer development.

According to the docking results, studied Les-6614 may interact with the LLP with the one amino acid (Ser150) from the catalytic triad consisting of Asp372, His429, and Ser150 [32] by the oxygen from the rhodanine ring (Figure 2). Additionally, the carboxyl group forms charge interaction with the His429. Tyr247 interplays with the molecule by Pi–sulfur and Pi–Pi stacked non-covalent interaction. Pro301, Cys60, and Cys334 residues attach to the indole ring and 1,3-thiazole ring by Pi–Alkyl.

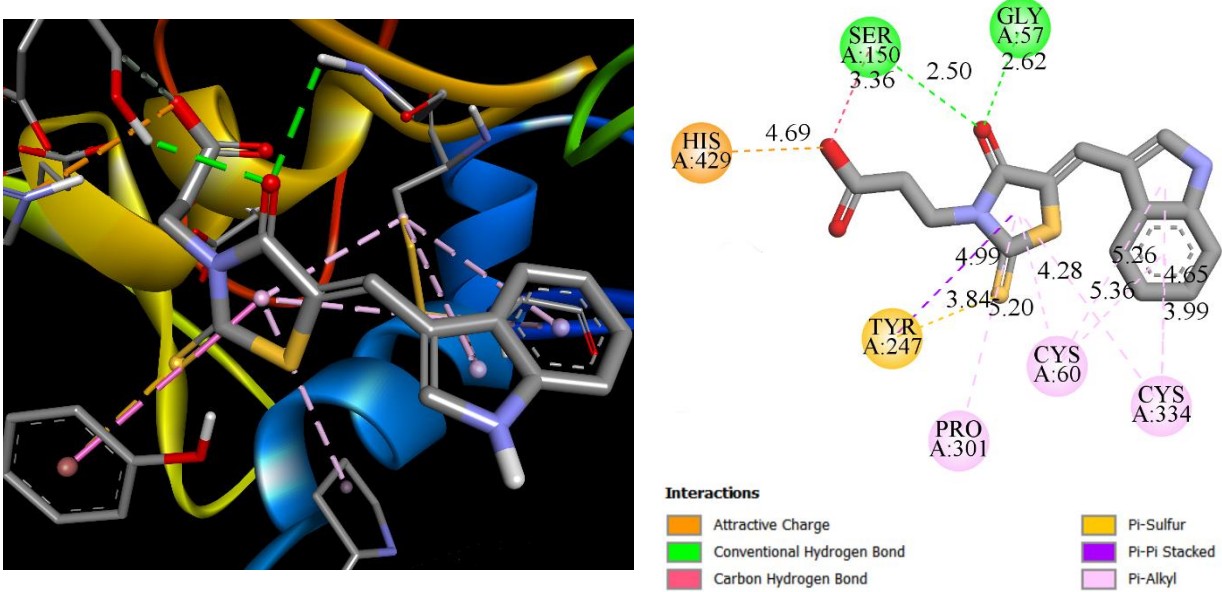

**Figure 2.** The predicted binding mode for the studied Les-6614 with the LLP (PDB 4CIA).

The compound may bind to the LLP enzyme by the two hydrogen bonds with the Trp132 (2.05 Å) and Cys479 (3.69 Å) (Figure 3). In addition, rhodanine and indole cores are launched by the number of hydrophobic interactions inside the cavity, formed by the lipophilic amino acids (Pi–Sigma, Pi–Sulfur, Pi–Pi (T-shaped), and Pi–Alkyl). Three Arginine amino acids form attractive charge and salt bridge interaction with the carboxyl group. Nevertheless, Les-6614 does not interact with the amino acid Arg374, which is

significant for inhibition, instead of the other reported competitive inhibitors dazoxiben and ozagrel [33].

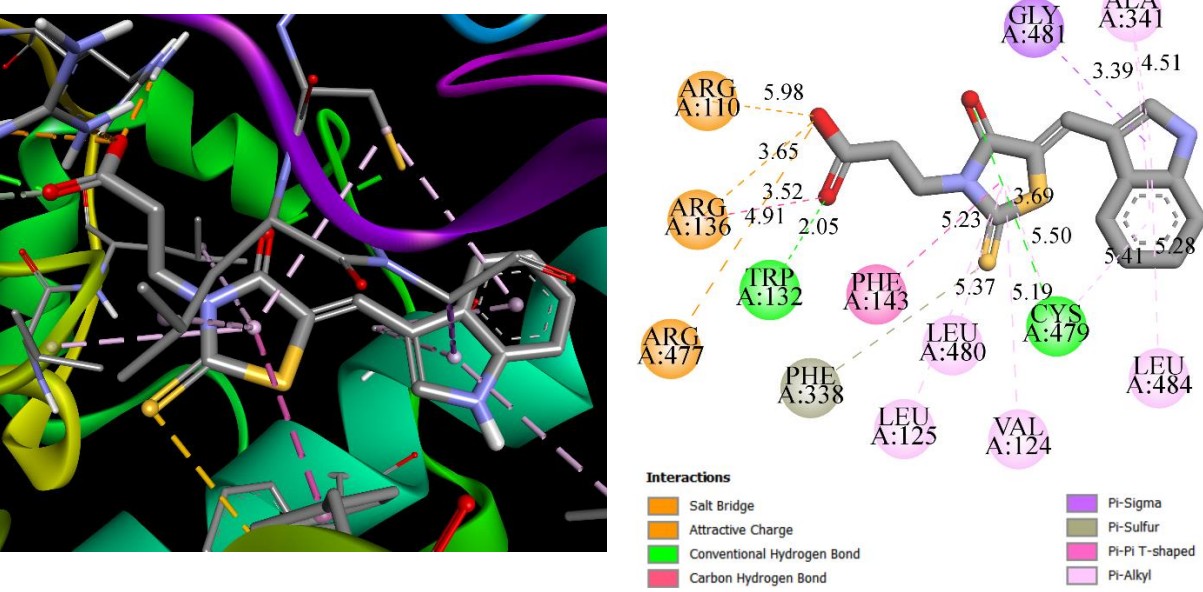

**Figure 3.** The predicted binding mode for the Les-6614 with the homology model of the TXAS.

It is worth noting that the studied compound in molecular docking results shows potential affinity to LLP and TXAS, which does not correlate with the above-mentioned results of SwissADME target predictions. These results can be explained by understanding the structure of the studied compound, which belongs to the rhodanine derivatives characterized by multitarget action, described in numerous data from the literature [34–36].

## 5. Conclusions

The selected compound Les-6614 was screened for several activities (antimicrobial, antifungal, cytotoxic, antitumor) and screened using molecular docking and the SwissAdme online server. The results show different activities of the studied rhodanine derivative, which can be interpreted as a moderately active "single agent on multiple targets for single/multiple disease/es". Special attention is drawn to the anti-allergic effect of the investigated compound as a potential PPARγ modulator. Furthermore, the molecular docking predicted that the Les-6614 might have possible anticancer properties and affinity for LLP and TXAS oncogenic targets. Considering the above, the tested 3-[5-(1*H*-indol-3-ylmethylene)-4-oxo-2-thioxothiazolidin-3-yl]-propionic acid is justified as a fruitful template for the search for a new class of multitarget therapeutic agents.

**Supplementary Materials:** The following supporting information can be downloaded at: https://www.mdpi.com/article/10.3390/scipharm91010013/s1, Figure S1. Similarity search on similar drug candidates or known bioactive molecules for the tested compound. Figure S2. Target prediction for a tested compound using SwissTargetPrediction.

**Author Contributions:** Contributions of 1st and 2nd authors are equal, Conceptualization, Y.K. and R.L.; Methodology and experimental work, Y.K., A.L., I.I. and O.H.; Data Analysis, Y.K., A.L., D.K. and R.L.; writing—review and editing, Y.K., A.L. and R.L.; Software, T.D., S.P., U.P. and I.P.; Project administration and Supervision, R.L. All authors have read and agreed to the published version of the manuscript.

**Funding:** The research leading to these results has received funding from the Ministry of Health of Ukraine, under the project numbers: 0121U100690, 0120U102460, 0123U100153, and the National Research Foundation of Ukraine, under the project number: 2020.02/0035.

**Institutional Review Board Statement:** The permission to conduct experiments on animals and isolate/work with cultures of microorganisms was approved by the protocol No. 6 of the commission on ethics of scientific research, experimental development and scientific works of Danylo Halytsky LNMU, Ukraine, from 25 June 2018.

**Informed Consent Statement:** Not applicable.

**Data Availability Statement:** The data presented in this study are available in this article.

**Acknowledgments:** The authors thank all the brave defenders of Ukraine who made the finalization of this article possible.

**Conflicts of Interest:** The authors declare no conflict of interest.

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
