# Peer review of "3-[5-(1H-Indol-3-ylmethylene)-4-oxo-2-thioxothiazolidin-3-yl]-propionic Acid as a Potential Polypharmacological Agent"

_scipharm, doi:10.3390/scipharm91010013_

Round 1

Reviewer 1 Report

In the manuscript entitled "3-[5-(1H-Indol-3-ylmethylene)-4-oxo-2-thioxo-thiazolidin-3-yl]-propionic acid as potential polypharmacological agent", the authors present an array of experiments to test the medical uses of their novel rhodamine derivative. The amount of work is significant and deserving of publication. The presentation of the work, however, requires improvement prior to publication.

First, significant English editing needs to be performed on the manuscript in order to increase readability. The consistent use of poor grammar and informal phrases detract from the message of the manuscript.  Some examples include:

- In line 31, the word “affine” is incorrectly used. Do the authors mean “has affinity for”?

- In line 33, the phrase “that studied compound was good bonded with” is poor English grammar. An article is missing (“that the studied compound”) and “good” should be replaced with its adverb version “well”.

- In line 157, “the absence 3D structures” is not correct and should be replaced with “the absent 3D structures” – or better yet rephrased so as to not have “because” as the first word of the sentence.

Further, all sections require editing to properly explain the work that was performed.

In the Introduction –

- While broadly applicable to the long-range goals of the group, the introduction does not properly introduce the work discussed in the rest of the manuscript. Additional transition to the scope of the work presented in the manuscript is requested.

- In line 63, previous work by the authors is mentioned but no citation for this work is provided. Please add the appropriate citation in the appropriate location(s).

- The last sentence or two of the introduction should summarize the work that is presented in the manuscript. Please add this summary.  (e.g., What does the current work adds to the previous results cited here?)

In the Materials and Methods –

- The sources of materials are not consistently cited.  Additionally, computer programs should have not only the company cited but also the version that was used to generate the images used in the manuscript.

- Please standardize the abbreviations for units used. For example, in line 86 μL is used but in line 88 mkL is used.

- The description of where microbes were obtained is unclear. Please clarify what was purchased (and where), and what samples were obtained from a primary source. Please include any applicable ethics statements required for primary patient samples.

- Please briefly summarize the guidelines that are cited in lines 123 and 135. Or, if you have already done so, please indicate that the subsequent text is as such.

- Abbreviations should be consistently defined at their first use. For example, ADMET in line 141. Currently, some basic abbreviations are defined while others are not. Please be consistent with what you do or do not consider worthy of definition.

- In section 2.5 about the methods used for the Molecular Docking studies, the introductory sentences that outline a rationale for its use are not appropriate for a Materials and Methods section. This level of information should belong with the description of the work in the Results and/or Discussion.

In the Results and Discussion sections –

- The subsections for the Results are all mis-numbered as 2.x rather than 3.x.

- In the opening sentences of section 3.1, how is the cited work different from the work presented in this manuscript? As presented, these cited works raise questions about the novelty of the current manuscript. While potentially invalid, this connotation should be addressed in order to quash it completely in the eyes of reviewers and readers.

- In lines 217-218, the authors use the caveat “in the tested concentrations” in regard to why their compound did not show the anticipated level of antitumor effect. Please describe how the tested concentrations were chosen and why additional concentrations were not/could not be tested. Again, the authors need to remove the negative connotations associated with their choice of wording.

- Figures 2 and 3 should be carefully examined. Perhaps including them in a supplemental text rather than the main text would be more appropriate. Figure 3 especially appears to be a screen-grab rather than primary data. The software used to create Figure 2 is not cited in the figure legend. What are the icons shown in Figure 2? For both figures: What version of the software was used? Are any citations needed for the software?

- The inclusion of molecular docking studies is tricky in this type of manuscript. First, the results must be stated as “predicted” rather than as absolutes until confirmatory binding studies are performed. Second, due to their predictive nature, they more likely should be included in the Discussion section than the Results section. This information is important to include, but the location of its inclusion should be carefully chosen.

- In Figures 4 and 5, the protein abbreviations are not defined. In fact, the abbreviations are not defined until the Discussion section, despite their use in the Results section. Moving this topic to the Discussion section would solve this discrepancy… Further, in regard to these two figures, more comprehensive figure legends are needed. What software was used to generate the images? Please include versions. Are all the colored interactions in Figure 4 used? The colors are so similar that distinguishing some of the pairs is almost impossible.

- In lines 296-297, the anticancer effect of the compound is called “weak”. Previously, this activity was shown to be not statistically significant (lines 217-220). Therefore, stating “weak” anticancer activity is not appropriate.

At this time, the manuscript is not quite ready for publication. The amount of information is appropriate, and no additional experiments likely need to be performed. The improvements requested are based solely in the communication of the data.

Author Response

Dear reviewer!

Many thanks for Your time spending and efforts in reviewing the manuscript.

We would like to comment on the main points.

"First, significant English editing needs to be performed on the manuscript in order to increase readability. The consistent use of poor grammar and informal phrases detract from the message of the manuscript."

We checked the English grammar and spelling of the manuscript according to Your notes.

"- In line 31, the word "affine" is incorrectly used. Do the authors mean "has affinity for"?"

Corrected as suggested.

"- In line 33, the phrase "that studied compound was good bonded with" is poor English grammar. An article is missing ("that the studied compound") and "good" should be replaced with its adverb version "well"."

Corrected as suggested.

"- In line 157, "the absence 3D structures" is not correct and should be replaced with "the absent 3D structures" – or better yet rephrased so as to not have "because" as the first word of the sentence."

Corrected as suggested.

"In the Introduction – - While broadly applicable to the long-range goals of the group, the introduction does not properly introduce the work discussed in the rest of the manuscript. Additional transition to the scope of the work presented in the manuscript is requested."

We completely agree with your comment. Changed the introduction a bit and added a few sentences to better explain the transition.

"- In line 63, previous work by the authors is mentioned but no citation for this work is provided. Please add the appropriate citation in the appropriate location(s)."

Citations to previous works were available in the manuscript. The citation has been moved to a more appropriate location.

"- The last sentence or two of the introduction should summarize the work that is presented in the manuscript. Please add this summary. (e.g., What does the current work adds to the previous results cited here?)"

We completely agree with your comment. Therefore, we added a few sentences about the current manuscript.

"In the Materials and Methods –- The sources of materials are not consistently cited. Additionally, computer programs should have not only the company cited but also the version that was used to generate the images used in the manuscript."

We insert sources of materials and also versions of software according to Your notes.

"- Please standardize the abbreviations for units used. For example, in line 86 μL is used but in line 88 mkL is used."

Corrected as suggested.

"- The description of where microbes were obtained is unclear. Please clarify what was purchased (and where), and what samples were obtained from a primary source. Please include any applicable ethics statements required for primary patient samples."

Information about the ethics commission is available in the "Institutional Review Board Statement" section. Permit to work with animals and to collect biomaterial from patients under one number and date.

"- Please briefly summarize the guidelines that are cited in lines 123 and 135. Or, if you have already done so, please indicate that the subsequent text is as such."

Corrected as suggested. Therefore, we have shortened this section.

"- Abbreviations should be consistently defined at their first use. For example, ADMET in line 141. Currently, some basic abbreviations are defined while others are not. Please be consistent with what you do or do not consider worthy of definition."

Corrected as suggested.

"- In section 2.5 about the methods used for the Molecular Docking studies, the introductory sentences that outline a rationale for its use are not appropriate for a Materials and Methods section. This level of information should belong with the description of the work in the Results and/or Discussion."

Yes. We agree that the first sentence might not be suitable for the materials and methods, and we inserted it in the results section.

In the Results and Discussion sections – - The subsections for the Results are all mis-numbered as 2.x rather than 3.x.

Corrected as suggested.

"In the opening sentences of section 3.1, how is the cited work different from the work presented in this manuscript? As presented, these cited works raise questions about the novelty of the current manuscript. While potentially invalid, this connotation should be addressed in order to quash it completely in the eyes of reviewers and readers."

Yes, we agree with the comments. Sentences that are not relevant to this study have been deleted.

"- In lines 217-218, the authors use the caveat "in the tested concentrations" in regard to why their compound did not show the anticipated level of antitumor effect. Please describe how the tested concentrations were chosen and why additional concentrations were not/could not be tested. Again, the authors need to remove the negative connotations associated with their choice of wording."

Yes, we agree with the comments. Therefore, we have removed this phrase.

"- Figures 2 and 3 should be carefully examined. Perhaps including them in a supplemental text rather than the main text would be more appropriate. Figure 3 especially appears to be a screen-grab rather than primary data. The software used to create Figure 2 is not cited in the figure legend. What are the icons shown in Figure 2? For both figures: What version of the software was used? Are any citations needed for the software?"

We moved Figures 2 and 3 to supplementary materials according to Your suggestions. We used an online platform SwissADME that is not software.

"- The inclusion of molecular docking studies is tricky in this type of manuscript. First, the results must be stated as "predicted" rather than as absolutes until confirmatory binding studies are performed. Second, due to their predictive nature, they more likely should be included in the Discussion section than the Results section. This information is important to include, but the location of its inclusion should be carefully chosen."

According to Your notes, we included some information from the molecular docking part from the results section to the discussion section.

"- In Figures 4 and 5, the protein abbreviations are not defined. In fact, the abbreviations are not defined until the Discussion section, despite their use in the Results section. Moving this topic to the Discussion section would solve this discrepancy… Further, in regard to these two figures, more comprehensive figure legends are needed. What software was used to generate the images? Please include versions. Are all the colored interactions in Figure 4 used? The colors are so similar that distinguishing some of the pairs is almost impossible."

According to Your notes, we included information from the molecular docking part from the results section to the discussion section and added a version of software used in this investigation.

"- In lines 296-297, the anticancer effect of the compound is called "weak". Previously, this activity was shown to be not statistically significant (lines 217-220). Therefore, stating "weak" anticancer activity is not appropriate."

We have rephrased it according to Your comments.

Reviewer 2 Report

This is interesting manuscript from a collaboration of largely Ukranian scientists. It details a wide range of biological activities and computational binding predictions on the title compound, which they proposed to be of potential use to modulate a range of targets relevant to a number of therapeutic areas. A number of corrections are requested prior to publication:

Abstract. English needs improving. Eg replace ‘is potentially affine’  for ‘ has similar affinities’ and ‘was good bonded’ for ‘showed good bonding’.

Figs 4 and 5 The investigated compound will be a carboxylate anion (NOT a free carboxylic acid) at pH 7.4. The anion should have been docked, but the images suggest COOH. This docking should be repeated, as ionic interactions would be expected.

Spelling of SwissAdme throughout the manuscript.

Figure 1: Number structure as 1, then use throughout the manuscript. It is confusing to have the compound numbered as 3 throughout?

Line 56…delete the first ‘potential’

Table 1 title…….’synthesised compounds’ should be replaced with 1 or ‘3-[5-(1H-Indol-3-ylmethylene)-4-oxo-2-thioxo-thiazolidin-3-yl]-propionic acid

Table 1  12….Gram-positive bac-teria……have bacteria on its own line.

17. Candida does not need a fullstop

 In Table 1, what does 00 mean/ Presumably ‘no activity’ or have these not been measured? Why not 0?

Section 2.2:   ‘3’ (twice) should be replaced with ‘3-[5-(1H-Indol-3-ylmethylene)-4-oxo-2-thioxo-thiazolidin-3-yl]-propionic acid or 1?

Table number should be 2 not 5.

What is the error for the J774.2 cell line?

In the Table, Commas should be replaced with full stops. Eg 0.57±1.22. and elsewhere.

Table 3…….give the name of the compound or 1 (not 3).

Line 252…..in silico in italics.

Lines 259-260 most capital letters not needed.

Line 284, 286, 294, 336……change 3 for compound name or 1?

Table 5 Legend. Need to refer to the Ki values….Are these measured or predicted?

Line 291…delete ‘literally’

Line 320. Not sure of the meaning of ‘with a distant pronounced effect on a single target.’

Line 358 activities (no letters in italics).

Line 229 alpha needs to be a symbol

Table 3……Compound 3?   & footnote not required? Not sure that I understand columns 3,4,6,7? This needs to be clearer.

Author Response

Dear Reviewer!

We would like to thank You for the revision and constructive comments that helped significantly improve the manuscript. Your suggestions have been incorporated in the revised manuscript (green highlight).

We would like to comment on the main points.

"Abstract. English needs improving. Eg replace 'is potentially affine' for 'has similar affinities' and 'was good bonded' for 'showed good bonding'."

Corrected as suggested.

"Figs 4 and 5 The investigated compound will be a carboxylate anion (NOT a free carboxylic acid) at pH 7.4. The anion should have been docked, but the images suggest COOH. This docking should be repeated, as ionic interactions would be expected."

Docking results were carried out for the tested compound in carboxylic acid form despite of level of PH. Considering that this compound is a weak acid and slightly dissociates, therefore, conducting docking studies for the anion form of this compound is not reasonably likely for strong electrolytes. In addition,  all physicochemical parameters of the studied compound were considered during the molecular docking process.

"Spelling of SwissAdme throughout the manuscript."

Corrected as suggested.

"Figure 1: Number structure as 1, then use throughout the manuscript. It is confusing to have the compound numbered as 3 throughout?"

Corrected as suggested.

"Line 56…delete the first ‘potential’”

Corrected as suggested.

“Table 1 title…….’synthesised compounds’ should be replaced with 1 or ‘3-[5-(1H-Indol-3-ylmethylene)-4-oxo-2-thioxo-thiazolidin-3-yl]-propionic acid”

It has been fixed according to your comment.

“Table 1  12….Gram-positive bac-teria……have bacteria on its own line.”

Corrected as suggested.

“In Table 1, what does 00 mean/ Presumably ‘no activity’ or have these not been measured? Why not 0?”

Corrected as suggested.

“Section 2.2:   ‘3’ (twice) should be replaced with ‘3-[5-(1H-Indol-3-ylmethylene)-4-oxo-2-thioxo-thiazolidin-3-yl]-propionic acid or 1?”

Corrected as suggested.

“Table number should be 2 not 5.”

Corrected as suggested.

“What is the error for the J774.2 cell line?”

Corrected as suggested.

“In the Table, Commas should be replaced with full stops. Eg 0.57±1.22. and elsewhere.”

Corrected as suggested.

“Table 3…….give the name of the compound or 1 (not 3).”

Corrected as suggested.

“Line 252…..in silico in italics.”

Corrected as suggested.

“Lines 259-260 most capital letters not needed.”

Corrected as suggested.

“Line 284, 286, 294, 336……change 3 for compound name or 1?”

Corrected as suggested.

“Table 5 Legend. Need to refer to the Ki values….Are these measured or predicted?”

All these Ki values are predicted. In addition, Autodock tools allow for predicting the Ki value of the docked ligand.

“Line 291…delete ‘literally’”

Corrected as suggested.

“Line 320. Not sure of the meaning of ‘with a distant pronounced effect on a single target.’”

Corrected as suggested.

“Line 358 activities (no letters in italics).”

Corrected as suggested.

“Line 229 alpha needs to be a symbol”

Corrected as suggested.

“Table 3……Compound 3?   & footnote not required? Not sure that I understand columns 3,4,6,7? This needs to be clearer.”

Corrected as suggested.

Sincerely,

Yulian Konechnyi

Reviewer 3 Report

In this present manuscript, Konechnyi et al. aimed to provide a succinct, comprehensive investigation entitled " 3-[5-(1H-Indol-3-ylmethylene)-4-oxo-2-thioxo-thiazolidin-3-yl]-2 propionic acid as potential polypharmacological agent" in the field of polypharmacology. Overall, the manuscript looks good and well written. The manuscript seems excellent all around. Although there are a few mistakes that need to be fixed before publication.

Minor Revision

The authors represent the polypharmacological role of the synthesized compound 3-[5-(1H-Indol-3-ylmethylene)-4-oxo-2-thioxo-thiazolidin-3-yl]-2 propionic acid and predict ADME for the molecule with the SwisAdme online server. It is worth checking in vitro pharmacological parameters (at least in vitro Caco-2 permeability and microsomal stability check) to rationalize the prediction.

1.      Write IgA instead of Ig A in the line 29 of page 1

2.      Write IgE instead of Ig E in the line 29 of page 1

3.      Review the table 2.  Provide the proper IC50 values of them. As its confusing to put a “Comma” instead of a “Decimal”. Similar query for page 6, line 220. For J774.2 cell line mention the SD.

4.      If 3-[5-(1H-Indol-3-ylmethylene)-4-oxo-2-thioxo-thiazolidin-3-yl]-2 propionic acid is the compound 3 then authors are advised to mention it before page 4, line 173. Would be good if the Compound 3 is mentioned in the Figure 1 which would be easy for the readers to understand the structure of the studied molecule in advance.

5.      Maintain uniformity of the reference section. In the reference 6 and 15 doi is missing.

Author Response

Dear Reviewer!

We would like to thank You for the revision and constructive comments that helped significantly improve the manuscript. Your suggestions have been incorporated in the revised manuscript (green highlight).

Dear reviewer!

Many thanks for Your time spending and efforts to review the manuscript.

We would like to comment the main points.

“1. Write IgA instead of Ig A in the line 29 of page 1”

Corrected as suggested.

“2. Write IgE instead of Ig E in the line 29 of page 1”

Corrected as suggested.

“Review the table 2. Provide the proper IC50 values of them. As its confusing to put a “Comma” instead of a “Decimal”. Similar query for page 6, line 220. For J774.2 cell line mention the SD.”

Corrected as suggested.

“If 3-[5-(1H-Indol-3-ylmethylene)-4-oxo-2-thioxo-thiazolidin-3-yl]-2 propionic acid is the compound 3 then authors are advised to mention it before page 4, line 173. Would be good if the Compound 3 is mentioned in the Figure 1 which would be easy for the readers to understand the structure of the studied molecule in advance.”

Corrected as suggested.

“Maintain uniformity of the reference section. In the reference 6 and 15 doi is missing.”

N 6 is a web page, added URL link; N 15 doesn`t have doi

Sincerely,

Yulian Konechnyi

Round 2

Reviewer 2 Report

Requested corrections have largely been completed. 

In Table 2 title, I just have a query about the units.....I think this should be microM (not microM/mL)?

Still a query about Figure 3 and ionisation. The pKa will be 3-5 (did you calculate/measure the pKa?), therefore expect a carboxylate anion for Le3-6614, that presumably would interact well with arginine (as shown for other TXAS inhibitors). But understand if the repeat of this modelling is difficult.

Author Response

Dear Reviewer!

We would like to thank You for the revision and constructive comments that helped significantly improve the manuscript. Your suggestions have been incorporated in the revised manuscript (red highlight).

“In Table 2 title, I just have a query about the units.....I think this should be microM (not microM/mL)?”

Thank  you for the comment, yes, you are right, it should be microM (μM), we corrected it.

“Still a query about Figure 3 and ionisation. The pKa will be 3-5 (did you calculate/measure the pKa?), therefore expect a carboxylate anion for Le3-6614, that presumably would interact well with arginine (as shown for other TXAS inhibitors). But understand if the repeat of this modelling is difficult.”

We agree that using ionic form of the compound would be more accurate for docking simulation. Therefore we performed docking one more. But for energy minimization and adding hydrogen we used Avogadro, because this program possesses the adding hydrogen to ligands according to different pH, instead of HyperChem.  The same we made for Dazoxiben owing to this reference ligand has the carboxyl group on its structure.  We can say that the predicted positions of the ligands inside the enzyme are completely the same ( as proof we added screen of the overlying positions of the compound in usual and ionic form with the LPP),  but we observed a slight increase of the binding energies. We can say that conclusions from the docking simulations remain the same, however, we thank you for your rational remark.   

(photo in PDF)

Sincerely,

Yulian Konechnyi
